

# Non-Markovian noise that cannot be
# dynamically decoupled by periodic spin echo pulses

**Daniel Burgarth**[1*]**, Paolo Facchi**[2,3]**, Martin Fraas**[4] **and Robin Hillier**[5]

**1** Center for Engineered Quantum Systems, Dept. of Physics & Astronomy,
Macquarie University, 2109 NSW, Australia
**2** Dipartimento di Fisica and MECENAS, Università di Bari, I-70126 Bari, Italy
**3** INFN, Sezione di Bari, I-70126 Bari, Italy
**4** Department of Mathematics, Virginia Tech, US
**5** Department of Mathematics and Statistics, Lancaster University, Lancaster LA1 4YF, UK

⋆ daniel.burgarth@mq.edu.au

## Abstract

**Dynamical decoupling is the leading technique to remove unwanted interactions in a vast range of quantum systems through fast rotations. But what determines the time-scale of such rotations in order to achieve good decoupling? By providing an explicit counterexample of a qubit coupled to a charged particle and magnetic monopole, we show that such time-scales cannot be decided by the decay profile induced by the noise: even though the system shows a quadratic decay (a Zeno region revealing non-Markovian noise), it cannot be decoupled by periodic spin echo pulses, no matter how fast the rotations.**

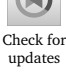
# 1   Introduction

Dynamical decoupling (DD) [1–3] is a generalisation of the famous Hahn spin echo in Nuclear Magnetic Resonance (NMR) [4]. It provides an intriguing control method to remove unwanted detrimental interactions in quantum technology and NMR. As a hardware-near error correction method, it has led to many experimental breakthroughs in the quantum realm [5–8]. This makes DD a key technique for quantum technology, where noise remains the main obstacle. But what type of noise can be removed by decoupling?

The physical intuition for decoupling is that active rotations of a quantum system, on a time-scale on which its interaction with an unwanted environment is effectively constant, leads to an averaging over the direction into which this interaction pushes the system. Which type of noise is amenable to DD is then related to the existence and experimental feasibility of such a time-scale. A commonly found physical intuition is that the decay profile (see Figure 1) of the unperturbed system dynamics is key: if the decay starts exponentially, it is too fast and cannot be decoupled, but if it starts with a quadratic 'Zeno' region, and thus reveals a degree of 'non-Markovianity' [1] of the bath, it can.

In the literature, there are several attempts to provide evidence to this simple picture. Commonly one considers exactly solvable dephasing models with Gaussian statistics [3] where an object known as the *bath spectral density* is closely linked to the decay shape. If it has a cut-off frequency, then the inverse of such cut-off determines the width of a Zeno region in the decay, and is an upper bound to a sufficiently small decoupling time to stabilise the decay up to first order. Optimal decoupling schemes can then be engineered for given spectral densities [3, 6]. Even for more complex environments bath spectral densities are shown to be useful in a perturbative regime [9, 10]. Further confidence in this picture came from the analogy with the quantum Zeno effect which was shown to be strictly related to decoupling [11]. While it was shown also that non-Markovianity can be detrimental to decoupling in a non-asymptotic regime, in analogy to the anti-Zeno effect, the Zeno region was deemed both necessary and sufficient for the asymptotic regime where the decoupling operations become fast enough [12–16].

Although such conjecture is based on a solid physical intuition, we were unable to find a rigorous proof. Indeed, our initial motivation was to provide a mathematical proof of the physical insight. Roughly, the idea was that the Zeno decay profile is connected to the finiteness of energy (i.e. the initial wave function is in the domain of the Hamiltonian), and finite energy is connected to the geometry of the wave-function trajectory that leads to small errors in a DD cycle. To say the same analytically, the domains are connected to convergence of Trotter's theorem, which is closely related to successful decoupling [17]. But we encountered hurdles: the connection between decay profiles and domains is not simple; and the convergence of Trotter's theorem is not simply related to Hamiltonian domains. Indeed, instead of finding a proof, we ended up with the opposite: a counterexample.

Our work shows that *there cannot be a direct connection between the existence of a Zeno region and decoupling for quantum baths*. The example also shows, as a byproduct, that even in 1D, quantum magnetism can be interesting. Last, but not least, the example we provide is also instructive (and fun) as it highlights some common pitfalls of formal calculations with unbounded operators.

---

[1]We should remark that the notion of 'non-Markovianity' is used in many different ways in the literature. For a recent review, see [22]. In the context of dynamical decoupling, it appears that it is mainly used to denote a non-exponential decay shape with a Zeno region [13], which is the way we use the term here.

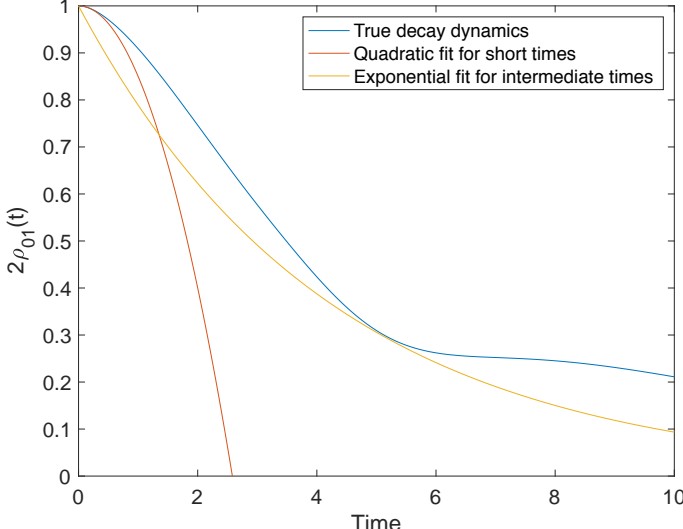

Figure 1: The model discussed in the paper induces decoherence of the off-diagonal elements of the qubit density matrix. The initial wave-function, chosen as $\psi_1 = \psi_2 \propto x e^{-|x|}$, is in the domain of the Hamiltonian (see Appendix). Therefore, the decay starts quadratically (as indicated by the fit for short times). This is contrasted with an exponential ('Markovian') fit.

## 2 Model

The model we have in mind is a single qubit coupled to a charged particle in 1D (see Figure 2). The interaction between the qubit and particle is spin-dependent: if the qubit is in state 0, the particle evolves freely; but if the state is 1, the particle feels an additional magnetic monopole at the origin. Because the particle evolution will depend on the spin state, decoherence is induced onto the qubit, and we show that it has a Zeno-like decay shape. Dynamical decoupling will quickly rotate the qubit, so that it should see the average of the particle evolution. The problem is, this average does not exist, because it would not conserve probability! As we will show, this sets a bound how much of the decoherence can be removed by decoupling, even in the limit of arbitrarily fast rotations.

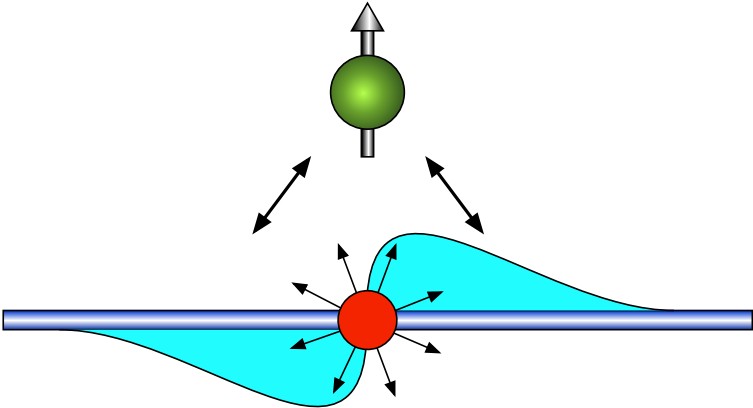

Figure 2: A spin-dependent interaction of a qubit (green) with a charged particle (cyan wave-function) in 1D and a monopole at the origin (red).

We first set the scene with an exactly solvable mathematical model for the bath, which we will later bring into physical context of the monopole. Consider the Hilbert space $\mathcal{H}_b = L^2(\mathbb{R}_+)$ of wave functions on the half-line $\mathbb{R}_+ = [0, +\infty)$. Let $p = -i\frac{d}{dx}$ be the momentum operator on $\mathcal{H}_b$ and define the family of self-adjoint operators

$$H_\alpha = p^2 + 2\alpha p\,, \tag{1}$$

with $\alpha \in \mathbb{R}$ and with common domain characterised by wave-functions that vanish at the origin (Dirichlet boundary condition):

$$D \equiv D(H_\alpha) = \{\psi \in H^2(\mathbb{R}_+),\ \psi(0) = 0\}\,, \tag{2}$$

where $H^2(\mathbb{R}_+)$ denotes the Sobolev space of twice weakly differentiable functions on $\mathbb{R}_+$. Further mathematical details are discussed in Appendix, in particular we show, by the method of images, that the time evolution is given by the unitaries

$$(U_\alpha(t)\psi)(x) = \frac{-i\,e^{i\alpha^2 t}}{\sqrt{\pi i t}} \int_0^{+\infty} e^{\frac{i(x^2 + y^2)}{4t}}\, e^{-i\alpha(x-y)} \sin\left(\frac{xy}{2t}\right) \psi(y)\mathrm{d}y\,. \tag{3}$$

Let us already mention here that the model is isomorphic to a non-relativistic particle on the *whole* line with charge $-\alpha$, subject to a magnetic Hamiltonian $(p + \alpha\,\mathrm{sgn}(x))^2$, with vector potential $A(x) = \mathrm{sgn}(x)$. This vector potential has a singularity (magnetic monopole) at the origin—a relic of the edge of the half-line—and is constant elsewhere.

## 3 Decoupling

With this model at hand, let us consider the coupled system-bath, where the system consists of a single qubit and the bath is given by the above model. Thus our Hilbert space is $\mathcal{H} = \mathbb{C}^2 \otimes \mathcal{H}_b = \mathbb{C}^2 \otimes L^2(\mathbb{R}_+)$ and we consider the total Hamiltonian

$$\hat{H} = -|0\rangle\langle 0| \otimes H_0 + |1\rangle\langle 1| \otimes H_\alpha = \begin{pmatrix} -H_0 & 0 \\ 0 & H_\alpha \end{pmatrix}, \tag{4}$$

which is self-adjoint on the domain $\mathbb{C}^2 \otimes D$. In order to formally decouple Hamiltonians of this structure, it suffices to consider the decoupling cycle $\{\mathbb{1}, X\}$, where $X$ is the Pauli matrix $X = \begin{pmatrix} 0 & 1 \\ 1 & 0 \end{pmatrix}$ acting on the qubit Hilbert space. After $n$ decoupling cycles of period $t/n$ an arbitrary initial state $\Psi = \begin{pmatrix} \psi_0 \\ \psi_1 \end{pmatrix}$ evolves into

$$\Psi^{(n)}(t) = \left(e^{-i\frac{t}{2n}X\hat{H}X}\, e^{-i\frac{t}{2n}\hat{H}}\right)^n \Psi, \tag{5}$$

where

$$X\hat{H}X = \begin{pmatrix} H_\alpha & 0 \\ 0 & -H_0 \end{pmatrix}. \tag{6}$$

We prove in Appendix that for any nontrivial $\psi_0, \psi_1 \in \mathcal{H}_b$ and $t > 0$, the Trotter limit $n \to \infty$ of (5) does *not* exist. The reason relies on the fact that (formally) one would get

$$\left(e^{-i\frac{t}{2n}H_\alpha}\, e^{i\frac{t}{2n}H_0}\right)^n \to e^{-i\frac{t}{2}(H_\alpha - H_0)} = e^{-it\alpha p}\,, \tag{7}$$

as $n \to \infty$, but $p$ is *not* a valid Hamiltonian on the half-line (it is not self-adjoint)! Indeed probability is not conserved, because $e^{-itp}\psi(x) = \psi(x - t)$, and for negative $t$ the wave-function is shifted to the left and eventually disappears from the positive half-line beyond the origin.

This is already interesting but not quite enough to conclude that DD does not work. Rather we have to study the time evolution of the reduced density matrix. In the absence of DD, at time $t \geq 0$ it is given by

$$\rho(t) = \begin{pmatrix} \|\psi_0\|^2 & \langle e^{-itH_\alpha}\psi_1 | e^{itH_0}\psi_0 \rangle \\ \langle e^{itH_0}\psi_0 | e^{-itH_\alpha}\psi_1 \rangle & \|\psi_1\|^2 \end{pmatrix}. \tag{8}$$

We can see that the system-bath coupling only induces dephasing.

In the presence of DD with $n$ bang-bang decoupling steps, this changes to $\rho^{(n)}(t)$, where the diagonal terms remain unchanged but the off-diagonal ones become

$$\rho_{01}^{(n)}(t) = \langle (e^{i\frac{t}{2n}H_0}e^{-i\frac{t}{2n}H_\alpha})^n \psi_1 | (e^{-i\frac{t}{2n}H_\alpha}e^{i\frac{t}{2n}H_0})^n \psi_0 \rangle. \tag{9}$$

*DD works if the decoupling operations switch off the unperturbed time evolution at any time $t > 0$ in the limit of infinitely fast decoupling pulses, namely if*

$$\rho^{(n)}(t) \to \rho(0), \quad \text{as} \quad n \to \infty. \tag{10}$$

We show in Appendix that for $\alpha > 0$, $\rho_{01}^{(n)}(t) \to \rho_{01}(0)$, so DD does work in this case. However, for $\alpha < 0$, we could not show such a convergence and there are in fact analytical indications that convergence does not hold in this case. We are going to prove this non-convergence numerically here, using a separable pure state $\begin{pmatrix} \psi_0 \\ \psi_1 \end{pmatrix} = \begin{pmatrix} \beta_0 \\ \beta_1 \end{pmatrix} \otimes \psi$. More precisely, we will show below that

$$\rho_{01}^{(n)}(t) \to \langle L(|\alpha|t)\psi_1 | L(|\alpha|t)\psi_0 \rangle = \overline{\beta}_1\beta_0 \int_{|\alpha|t}^\infty |\psi(x)|^2 \mathrm{d}x, \tag{11}$$

where $L(t)\psi(x) = \psi(x+t)\theta(x)$ is the left shift on the half-line, $\theta$ being the Heaviside step function. Since the right-hand side is in general different from $\rho_{01}(0) = \overline{\beta}_1\beta_0 \int_0^\infty |\psi(x)|^2 \mathrm{d}x$, this will imply that DD does not work in this model. On the other hand, one can show ([18], Appendix) that there is a Zeno region (see Figure 1). Thus we have *non-Markovian noise which cannot be dynamically decoupled*.

In order to prove (11) numerically, let us consider a 'cat state', $\beta_0 = \beta_1 = 1/\sqrt{2}$, on the system and the wave function $\psi(x) = \sqrt{4/3}\, x^2 \exp(-x) \in D$. We choose $\alpha = -2$ and $t = 1$ so that one should have $\rho_{01}(t) \to \int_2^\infty |\psi(x)|^2 \mathrm{d}x/2 = \frac{103}{6e^4} \approx 0.3144$. Numerically, we implement (3) with a suitable discretization and cut-off for $x$ and find a painfully slow convergence: for $n = 200$ we obtain 0.3066, for $n = 400$ it is 0.3090, and for $n = 800$ we get 0.3105. In particular, DD does not work for negative $\alpha$. On the other hand, for positive $\alpha = +2$ it works very well, and already after $n = 5$ operations we have $\rho_{01}^{(n)}(1) > 0.999/2$. We gained further confidence in the numerics by analyzing different initial states and different values for $t$ and $\alpha$ and finding a good agreement with the predicted value from Eq. (11) (Table 1).

## 4 Qualitative picture

Now that we have seen from a mathematical point of view why DD does not work, let us develop a qualitative and more physical understanding of why DD fails. For large $n$ the decoupling dynamics shifts the wave function to the left or to the right depending on the sign of $\alpha$. The part of the wave-function far away from 0 does not change its shape significantly, but the part that would have hit zero by pure shift has to be reflected and acquires a phase upon the reflection.

Table 1: Testing the limit numerically for a variety of initial wave functions and parameters $\alpha$. We chose a discretization of $\Delta x = 0.01$ and a cut-off of $x \leq 20$.

| Wavefunction | $2\alpha$ | $2\rho_{01}^{(n)}(t=2)@n=20$ | $\int_{|\alpha t|}^{\infty}|\psi(x)|^2 dx$ |
|---|---|---|---|
| $\propto x^2 e^{-x^2/5}$ | -2 | 0.62 | 0.67 |
| $\propto x^2 e^{-x^2/5}$ | -1 | 0.97 | 0.98 |
| $\propto x^2 e^{-x^2/4}$ | -2 | 0.50 | 0.53 |
| $\propto x^2 e^{-x^2/4}$ | -1 | 0.95 | 0.96 |
| $\propto x e^{-x^2/5}$ | -2 | 0.33 | 0.35 |
| $\propto x e^{-x^2/5}$ | -1 | 0.81 | 0.84 |
| $\propto x e^{-x/5}$ | -2 | 0.95 | 0.95 |
| $\propto x e^{-x/5}$ | -1 | 0.99 | 0.99 |

This can be explicitly demonstrated when the Dirichlet boundary condition (2) is replaced by a potential barrier $V(x)$. The Hamiltonian $H_\alpha = p^2 + V(x) + 2\alpha p$ then satisfies $[H_0, H_\alpha] = i2\alpha V'(x)$ and to the first order in $n$ the BCH formula gives (formally)

$$
\left(e^{-i\frac{t}{2n}H_\alpha} e^{i\frac{t}{2n}H_0}\right)^n \approx e^{-i\alpha t(p-\frac{1}{2}\frac{t}{n}V')},
$$
$$
\left(e^{-i\frac{t}{2n}H_0} e^{i\frac{t}{2n}H_\alpha}\right)^n \approx e^{-i\alpha t(p+\frac{1}{2}\frac{t}{n}V')}. \tag{12}
$$

The dynamics generated by the right hand side can be explicitly calculated and gives $\psi_k(x,t) \approx e^{(-1)^k i\frac{t}{2n}[V(x)-V(x-\alpha t)]}\psi(x-\alpha t)$, for $k = 0,1$. For the potential barrier $V(x) = V$ for $x < 0$ and $V(x) = 0$ for $x \geq 0$ we get $\psi_k(x,t) = 0$ for $x \leq \alpha t$ and

$$
\psi_k(x,t) \approx \begin{cases} \psi(x-\alpha t) & x \geq 0, \\ e^{(-1)^k i\frac{t}{2n}V}\psi(x-\alpha t) & 0 \geq x \geq \alpha t. \end{cases} \tag{13}
$$

For $\alpha > 0$ the second option is empty and the evolution shifts the wave function to the right. For $\alpha < 0$ the wave function is shifted to the left and the part of it that reaches the barrier gets a phase factor. The phase factor is relevant for $V > n$. For the off-diagonal element of the density matrix this gives

$$
\rho_{01}(t) \approx \overline{\beta}_1 \beta_0 \left(e^{i\frac{t}{n}V} \int_0^{|\alpha|t} |\psi(x)|^2 dx + \int_{|\alpha|t}^{\infty} |\psi(x)|^2 dx\right), \tag{14}
$$

and the acquired phase leads to the decay of the off-diagonal element of the density matrix.

In the above approximation the phase acquired upon passing the barrier is constant and subsequently the decay of the off-diagonal element is transient. By a work of Berry [19], for Dirichlet wall the reflected wave-function is expected to have a fractal shape and the reflected part of the wave-function then does not contribute to the off-diagonal element, leading to (11). This picture is well supported by numerics which show a fractal-like, highly oscillatory part of the wave function in the presence of DD (Figure 3), and an unbounded increase in its kinetic energy.

This qualitative picture might have implications for other systems, too. For example, to dynamically decouple an interacting bath prepared in a low-energy state, one should design the dynamical decoupling such that it would not cause the excitations of high-energy modes or the decoupling scheme could be spoiled.

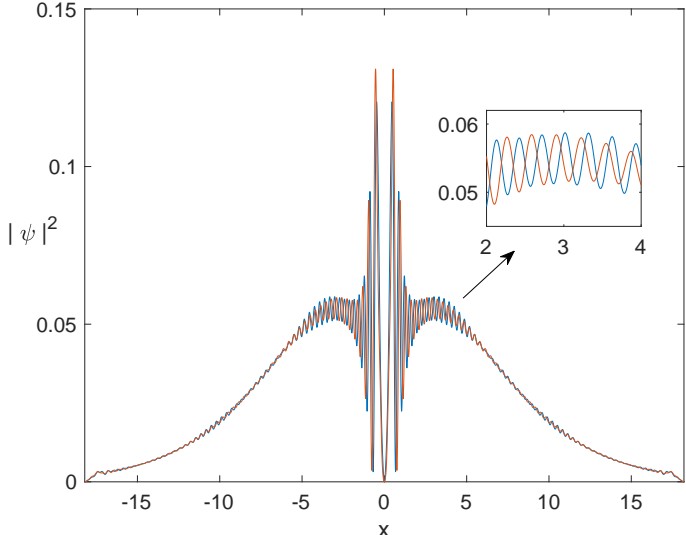

Figure 3: The two components of the wave-function for $n = 20$ decoupling steps. We plot the square modulus for the monopole picture on the whole real line. The inset shows the phase differences leading to an overlap of $\approx 0.95$. The initial-wave function was chosen as $\psi_0(x) = \psi_1(x) \propto x e^{-|x|/5}$.

## 5 Conclusion

At this point the reader might wonder how physical an environment on the half-line is. In response, we show in Appendix that the model is equivalent to the aforementioned physical picture of a magnetic particle on the full line. One can easily extend the model further to include several such environments to obtain something closer to a heat-bath. Addition of local system Hamiltonians and bounded bath Hamiltonians would also not change the non-convergence of Trotter [26], but would generally spoil the exact solvability of the model.

There are several striking conclusions to be drawn. Most importantly, we have shown that the physics mechanism, that a Zeno region provides a time-scale for which dynamical decoupling works, is incorrect in general. As our analysis showed, such picture is too simplistic, as it neglects the *back-action of the system onto the bath*. While for many systems encountered in practice such reasoning is probably valid, it cannot hold in general, because we obtained an explicit counterexample. Combined with our previous work which showed that certain models without Zeno region *can* be decoupled [20], it means that the Zeno region is neither necessary nor sufficent for noise suppression. From our perspective this provides further evidence [21, 22] that the relevant time-scales for DD are not something that one can see by looking at an unperturbed reduced system dynamics, but a property of the system *and* bath.

Dynamical decoupling works for positive $\alpha$ but not for negative one. Equivalently one can conclude that we have provided an example of a Hamiltonian $\hat{H}$ which cannot be decoupled but for which $-\hat{H}$ can be decoupled. This is particular striking because domains, bath state and correlations are the same for both cases.

Our analysis also highlights the intriguing subtleties of unbounded operators. On the half-line, the dynamics generated by the Hamiltonians $p^2$ and $p^2 + 2\alpha p$ do not commute; and $p$ is not even a valid Hamiltonian. This shows that the perturbative expansion of the time evolution in terms of a series fails. In the literature on open systems, the Hamiltonian is usually decomposed as $\hat{H} = H_S \otimes \mathbb{1} + \mathbb{1} \otimes H_B + H_{SB}$ with $\mathrm{tr}_S H_{SB} = 0$. Interestingly in our case $H_B = \alpha p$ is *not* a Hamiltonian, so the decomposition is not physical. Likewise, the interaction

picture of $H_{SB}$ with respect to $H_S$ and $H_B$ is undefined. The lack of perturbative expansion and such a decomposition implies that the bath spectrum [3] of the model cannot be defined, either. Finally our model shows that in the time-dependent case, not all dynamical effects of quantum magnetism on $L^2(\mathbb{R})$ can be removed by gauge transformations [23]. In future works, it will be interesting to work out under which additional hypothesis one can recover a simpler connection between initial decay and decoupling, and to study how realistic this hypothesis is in the lab.

**Funding information**    DB acknowledges support by the Australian Research Council (project number FT190100106). PF is partially supported by the Italian National Group of Mathematical Physics (GNFM-INdAM), by Istituto Nazionale di Fisica Nucleare (INFN) through the project "QUANTUM", and by Regione Puglia and QuantERA ERA-NET Cofund in Quantum Technologies (GA No. 731473), project PACE-IN.

## A    The model and its dynamics

Consider the Hilbert space $\mathcal{H}_b = L^2(\mathbb{R}_+)$ of wave functions on the half-line $\mathbb{R}_+ = [0, +\infty)$, and the dense linear subspace $\mathcal{D}(0, +\infty)$ of smooth functions with compact support in $(0, +\infty)$. The operators $p = -i\frac{d}{dx}$ and $p^2 = -\frac{d^2}{dx^2}$ are symmetric on $\mathcal{D}(0, +\infty)$, and so is their linear combination

$$H_\alpha = p^2 + 2\alpha p, \tag{15}$$

with $\alpha \in \mathbb{R}$. It is not difficult to show that the operators $H_\alpha$ are self-adjoint on the common domain

$$D \equiv D(H_\alpha) = \{\psi \in H^2(\mathbb{R}_+), \, \psi(0) = 0\}. \tag{16}$$

Indeed, for $\psi, \varphi \in H^2(\mathbb{R}_+)$, by integration by parts, one gets

$$\langle H_\alpha \varphi | \psi \rangle - \langle \varphi | H_\alpha \psi \rangle = \overline{\varphi(0)}\big(\psi'(0) + 2i\alpha\psi(0)\big) - \overline{\varphi'(0)}\psi(0). \tag{17}$$

The crucial point is that $H_\alpha - H_\beta = 2(\alpha - \beta)p$ is symmetric but not self-adjoint for $\alpha \neq \beta$, so it does not generate a unitary evolution.

Next, let us explicitly construct the unitary groups $e^{-itH_\alpha}$ by the method of images. We first notice the explicit unitary operator $W : L^2(\mathbb{R}_+) \to L^2_o(\mathbb{R})$ between $L^2(\mathbb{R}_+)$ and the Hilbert space $L^2_o(\mathbb{R})$ of odd square integrable functions on the whole line $\mathbb{R}$, defined by

$$\tilde{\psi}(x) = (W\psi)(x) = \frac{1}{\sqrt{2}} \, \text{sgn}(x)\psi(|x|), \tag{18}$$

for all $x \in \mathbb{R}$ [24]. Notice that $\tilde{\psi}(-x) = -\tilde{\psi}(x)$ is an odd function, and $\|\tilde{\psi}\| = \|W\psi\| = \|\psi\|$. The inverse unitary $W^\dagger$ is given by $(W^\dagger \tilde{\psi})(x) = \sqrt{2}\tilde{\psi}(x) = \psi(x)$, for all $x \in \mathbb{R}_+$.

Recall that on $L^2(\mathbb{R})$ the free particle evolves as

$$(e^{-itp^2}\tilde{\psi})(x) = \frac{1}{\sqrt{4\pi it}} \int_{\mathbb{R}} e^{\frac{i(x-y)^2}{4t}} \tilde{\psi}(y) dy, \quad t > 0, \tag{19}$$

valid for $\tilde{\psi} \in L^2(\mathbb{R}) \cap L^1(\mathbb{R})$, and on the full Hilbert space by density [25]. Thus on $L^2(\mathbb{R}_+)$,

we get for all $x \geq 0$ and $t > 0$:

$$
\begin{aligned}
(U_0(t)\psi)(x) &= (\sqrt{2}\,e^{-itp^2}\tilde{\psi})(x) \\
&= \frac{1}{\sqrt{4\pi it}}\int_0^{+\infty}\left[e^{\frac{i(x-y)^2}{4t}} - e^{\frac{i(x+y)^2}{4t}}\right]\psi(y)\mathrm{d}y \\
&= \frac{-i}{\sqrt{\pi it}}\int_0^{+\infty}e^{\frac{i(x^2+y^2)}{4t}}\sin\left(\frac{xy}{2t}\right)\psi(y)\mathrm{d}y.
\end{aligned}
\tag{20}
$$

Notice that $(U_0(t)\psi)(0) = 0$ for all $t$, since $p^2$ preserves parity. This is consistent with the fact that $U_0(t)D(H_0) = D(H_0)$.

As for $H_\alpha$ with $\alpha \neq 0$, notice that $e^{i\alpha x}D(H_\alpha) = D(H_\alpha)$ and

$$
p^2\,e^{i\alpha x}\,\psi(x) = e^{i\alpha x}\left(p^2 + 2\alpha p + \alpha^2\right)\psi(x),
\tag{21}
$$

for all $\psi \in D(H_\alpha)$. Therefore, $U_\alpha(t) = e^{i\alpha^2 t}\,e^{-i\alpha x}\,U_0(t)\,e^{i\alpha x}$, whence

$$
(U_\alpha(t)\psi)(x) = \frac{-i\,e^{i\alpha^2 t}}{\sqrt{\pi it}}\int_0^{+\infty}e^{\frac{i(x^2+y^2)}{4t}}\,e^{-i\alpha(x-y)}\sin\left(\frac{xy}{2t}\right)\psi(y)\mathrm{d}y.
\tag{22}
$$

Notice again that $U_\alpha(t)D(H_\alpha) = D(H_\alpha)$, as it should.

Translated to $L_o^2(\mathbb{R})$, we get

$$
\tilde{U}_\alpha(t) = WU_\alpha(t)W^\dagger = e^{i\alpha^2 t}\,e^{-i\alpha|x|}\,e^{-itp^2}\,e^{i\alpha|x|}\,.
\tag{23}
$$

But, for all $\tilde{\psi} \in W(D) = H_o^2(\mathbb{R})$, one gets

$$
e^{-i\alpha|x|}\,p^2\,e^{i\alpha|x|}\,\tilde{\psi} = \left(p + \alpha\,\mathrm{sgn}(x)\right)^2\tilde{\psi}\,,
\tag{24}
$$

and thus

$$
\tilde{U}_\alpha(t) = e^{i\alpha^2 t}\,e^{-it(p+\alpha\,\mathrm{sgn}(x))^2}\,.
\tag{25}
$$

Therefore, the unitary evolution $U_\alpha(t)$ on the half-line generated by the Hamiltonian $H_\alpha$ in (15) with Dirichlet boundary conditions, is unitarily equivalent to the unitary evolution (25) of odd wave functions on the full line. In the latter picture we have (up to a phase) the dynamics of a non-relativistic particle on a line with charge $-\alpha$, subject to a magnetic Hamiltonian $(p + \alpha\,\mathrm{sgn}(x))^2$, with vector potential $A(x) = \mathrm{sgn}(x)$. This vector potential has a singularity (monopole) at the origin—a relic of the edge of the half-line—and is constant elsewhere. And in this picture the non-commutativity of $p^2$ and $(p + \alpha\,\mathrm{sgn}(x))^2$ is apparent and obviously due to the monopole.

## B Dynamical decoupling computations

Before we can deal with the decoupling dynamics, we need some preparation. Let us consider Trotter-Kato dynamics where we imagine quickly switching between the evolution of $H_\alpha$ and $-H_0$ on $\mathcal{H}_b = L^2(\mathbb{R}_+)$. It is well known that the operator $p$ has no self-adjoint extensions on $L^2(\mathbb{R}_+)$. This follows e.g. from an adaptation of [26, Ex.1 in Sect.X.1] because the deficiency indices of $p$ are $(1,0)$. This is reflected in the properties of translations on the half-line, the left translation on $L^2(\mathbb{R}_+)$,

$$
(L(t)\psi)(x) = \psi(x+t)
\tag{26}
$$

and the right translation

$$(R(t)\psi)(x) = \begin{cases} \psi(x-t) & x \geq t, \\ 0 & x \leq t, \end{cases} \tag{27}$$

for $t \geq 0$. These translations are adjoints of each other namely $R(t) = L(t)^\dagger = R(t)^{\dagger\dagger}$, and $R(t)$ is an isometry namely $L(t)R(t)\psi = \psi$ and

$$(R(t)L(t)\psi)(x) = \begin{cases} \psi(x) & x \geq t, \\ 0 & x \leq t. \end{cases} \tag{28}$$

The right (left) translation is generated by the minimal (maximal) closed but not self-adjoint extension of the momentum [27, Ch.8], $p_{\min}$ ($p_{\max}$), i.e.

$$R(t) = e^{-itp_{\min}}, \quad L(t) = e^{itp_{\max}}. \tag{29}$$

We have $p_{\min} = p_{\max}^\dagger = p_{\min}^{\dagger\dagger}$ but by (28), $p_{\min} \neq p_{\max}$: the operators differ in their domain, $D(p_{\min}) = \{\psi \in H^1(\mathbb{R}_+), \psi(0) = 0\}$ and $D(p_{\max}) = H^1(\mathbb{R}_+)$, so $p_{\min}$ is symmetric while $p_{\max}$ is not.

In order to proceed, we need the following theorem due to Chernoff [28] (see [29, Ch.8, Theorem 5]). Recall that for operators $A, B$ with domains $D(A)$ and $D(B)$ the algebraic sum $A + B$ is defined on the domain $D(A) \cap D(B)$, and the closure of an operator $C$ is denoted by $\overline{C}$.

**Theorem.** *Let $A, B$ and $\overline{A+B}$ be generators of contraction semigroups. Then*

$$\left( e^{\frac{t}{n}A} e^{\frac{t}{n}B} \right)^n \psi \quad \rightarrow \quad e^{t(\overline{A+B})} \psi, \quad t \geq 0,$$

*as $n \to +\infty$, holds for all $\psi$.*

We use this theorem for $A = -\frac{i}{2}H_\alpha$ and $B = \frac{i}{2}H_0$ that have a common domain $D$ in (16). The formal algebraic sum $A + B = -i\alpha p$ then has the same domain, and hence we recognize that $\overline{A+B} = -i\alpha p_{\min}$. We have to distinguish the two cases $\alpha < 0$ and $\alpha > 0$.

For $\alpha > 0$, $\overline{A+B}$ is $\alpha$ times the generator of the right shift contraction semigroup, so

$$\left( e^{-i\frac{t}{2n}H_\alpha} e^{i\frac{t}{2n}H_0} \right)^n \psi \rightarrow R(\alpha t)\psi, \quad \text{as} \quad n \to \infty, \tag{30}$$

for any $t \geq 0$ and $\psi \in L^2(\mathbb{R}_+)$. We also note that for $A$ and $B$ exchanged the theorem gives

$$\left( e^{i\frac{t}{2n}H_0} e^{-i\frac{t}{2n}H_\alpha} \right)^n \psi \rightarrow R(\alpha t)\psi, \quad \text{as} \quad n \to \infty, \tag{31}$$

for $t \geq 0$ (and positive $\alpha$).

For $\alpha < 0$, the situation is different and more involved. First of all, by taking adjoints and recalling that $H_0, H_\alpha$ are self-adjoint, we have

$$\begin{aligned} \langle \phi | \left( e^{-i\frac{t}{2n}H_\alpha} e^{i\frac{t}{2n}H_0} \right)^n \psi \rangle &= \langle \left( e^{-i\frac{t}{2n}H_0} e^{i\frac{t}{2n}H_\alpha} \right)^n \phi | \psi \rangle \\ &\rightarrow \langle R(-\alpha t)\phi | \psi \rangle = \langle \phi | L(|\alpha|t)\psi \rangle, \end{aligned} \tag{32}$$

as $n \to \infty$, for all $\phi, \psi \in \mathcal{H}$ and $t \geq 0$. Hence if $\left( e^{-i\frac{t}{2n}H_\alpha} e^{i\frac{t}{2n}H_0} \right)^n$ converges in the strong sense, it has to converge to $L(|\alpha|t)$. However, for any $\psi \in \mathcal{H}$, the unitarity of $\left( e^{-i\frac{t}{2n}H_\alpha} e^{i\frac{t}{2n}H_0} \right)^n$ implies that

$$\begin{aligned} \| \left( e^{-i\frac{t}{2n}H_\alpha} e^{i\frac{t}{2n}H_0} \right)^n \psi - L(|\alpha|t)\psi \|^2 &= 1 + \|L(|\alpha|t)\psi\|^2 - 2\operatorname{Re}\langle \left( e^{-i\frac{t}{2n}H_\alpha} e^{i\frac{t}{2n}H_0} \right)^n \psi | L(|\alpha|t)\psi \rangle \\ &\rightarrow 1 + \|L(|\alpha|t)\psi\|^2 - 2\operatorname{Re}\langle L(|\alpha|t)\psi | L(|\alpha|t)\psi \rangle = 1 - \|L(|\alpha|t)\psi\|^2, \quad \text{as} \quad n \to \infty, \end{aligned} \tag{33}$$

where the penultimate line follows from (32); this is 0 if and only if $\psi$ has support in $[|\alpha|t, \infty)$. Thus we have strong convergence

$$\left( e^{-i\frac{t}{2n}H_\alpha} e^{i\frac{t}{2n}H_0} \right)^n \psi \to L(|\alpha|t)\psi, \quad \text{as} \quad n \to \infty, \tag{34}$$

if and only if $\psi$ has its support in $[|\alpha|t, \infty)$.

It is instructive to note that (30) cannot hold for $t \leq 0$. Suppose that $(e^{-i\frac{t}{2n}H_1} e^{i\frac{t}{2n}H_0})^n \psi$ converges for all $\psi$ and all $t$. Then it has to converge to a strongly continuous unitary one-parameter group $e^{-itC}$, with $2C$ being a self-adjoint extension of the algebraic sum $H_1 - H_0$. In this case, the sum is densely defined and given by $H_1 - H_0 = 2p$, and we conclude that $C$ is a self-adjoint extension of $p$. This is a contradiction to the fact that $p$ has no self-adjoint extension on the half line. Notice that this is also a proof of the counterintuitive but known fact that the evolutions generated by the Hamiltonian $p^2 + p$ and by $p^2$, respectively, do not commute.

We are now ready to turn to the actual dynamical decoupling computations. We consider the Hilbert space $\mathcal{H} = \mathbb{C}^2 \otimes L^2(\mathbb{R}_+)$ and the Hamiltonian

$$\hat{H} = -|0\rangle\langle 0| \otimes H_0 + |1\rangle\langle 1| \otimes H_\alpha = \begin{pmatrix} -p^2 & 0 \\ 0 & p^2 + 2\alpha p \end{pmatrix}. \tag{35}$$

By (16), this is self-adjoint on the domain $\mathbb{C}^2 \otimes D$. In order to formally decouple Hamiltonians of this structure, it suffices to consider the decoupling cycle $\{\mathbb{1}, X\}$, where $X$ is the Pauli matrix $X = \begin{pmatrix} 0 & 1 \\ 1 & 0 \end{pmatrix}$. We claim that for any $\psi_0, \psi_1 \in L^2(\mathbb{R}_+)$ the Trotter limit

$$\lim_{n \to \infty} \left( e^{-i\frac{t}{2n}X\hat{H}X} e^{-i\frac{t}{2n}\hat{H}} \right)^n \begin{pmatrix} \psi_0 \\ \psi_1 \end{pmatrix} \tag{36}$$

does not exist for all $t$.

By the definition of $\hat{H}$ we have

$$X\hat{H}X = \begin{pmatrix} p^2 + 2\alpha p & 0 \\ 0 & -p^2 \end{pmatrix} \tag{37}$$

and hence

$$\left( e^{-i\frac{t}{2n}X\hat{H}X} e^{-i\frac{t}{2n}\hat{H}} \right)^n \begin{pmatrix} \psi_0 \\ \psi_1 \end{pmatrix} = \begin{pmatrix} \left( e^{-i\frac{t}{2n}H_\alpha} e^{i\frac{t}{2n}H_0} \right)^n \psi_0 \\ \left( e^{i\frac{t}{2n}H_0} e^{-i\frac{t}{2n}H_\alpha} \right)^n \psi_1 \end{pmatrix}. \tag{38}$$

To see the impact this has on the dynamics, suppose that the system is in a separable state $\Psi = \begin{pmatrix} \psi_0 \\ \psi_1 \end{pmatrix} = \begin{pmatrix} \beta_0 \\ \beta_1 \end{pmatrix} \otimes \psi$, $\|\Psi\| = 1$, with the state of the bath $\psi$ in the domain $D$ and with $\int_{|\alpha|t}^\infty |\psi(x)|^2 dx < 1$ for some $\alpha$ and $t$.

The total initial state $\Psi$ is then in the domain of $\hat{H}$ and therefore displays a Zeno region of quadratic decay [18]. Indeed, one gets as $t \to 0$

$$\langle \Psi | e^{-it\hat{H}} \Psi \rangle = 1 - it\langle \Psi | \hat{H} \Psi \rangle - \frac{t^2}{2} \langle \hat{H}\Psi | \hat{H}\Psi \rangle + o(t^2), \tag{39}$$

whence

$$|\langle \Psi | e^{-it\hat{H}} \Psi \rangle|^2 = 1 - t^2 \left( \|\hat{H}\Psi\|^2 - \langle \Psi | \hat{H}\Psi \rangle^2 \right) + o(t^2). \tag{40}$$

We can conclude by (34) that the limit $n \to \infty$ of (38) does not exist. This does, however, not quite decide the fate of DD yet as the existence of the limit is a sufficient but not necessary condition in order for DD to work. Instead, we need to look at the reduced dynamics of

the system, and conclude that this does not converge to the identity evolution in the limit of $n \to \infty$. Without DD, the state will evolve under the Hamiltonian into $\begin{pmatrix} e^{itH_0}\psi_0 \\ e^{-itH_\alpha}\psi_1 \end{pmatrix}$. The reduced density matrix is given by

$$\rho(t) = \begin{pmatrix} \|\psi_0\|^2 & \langle e^{-itH_\alpha}\psi_1 | e^{itH_0}\psi_0 \rangle \\ \langle e^{itH_0}\psi_0 | e^{-itH_\alpha}\psi_1 \rangle & \|\psi_1\|^2 \end{pmatrix}. \tag{41}$$

We can see that the system-bath coupling only induces dephasing.

In the presence of DD at time $t > 0$ and with $n$ decoupling steps, the state evolves according to (38) and the reduced density matrix becomes $\rho^{(n)}(t)$, where the diagonal terms remain unchanged but the off-diagonal ones are

$$\rho_{01}^{(n)}(t) = \langle (e^{i\frac{t}{2n}H_0} e^{-i\frac{t}{2n}H_\alpha})^n \psi_1 | (e^{-i\frac{t}{2n}H_\alpha} e^{i\frac{t}{2n}H_0})^n \psi_0 \rangle = \overline{\rho_{10}^{(n)}(t)}. \tag{42}$$

DD works if the decoupling operations switch off the unperturbed time evolution at any time $t > 0$ in the limit of infinitely fast decoupling pulses, namely if

$$\rho^{(n)}(t) \to \rho(0), \quad \text{as} \quad n \to \infty. \tag{43}$$

Let us first consider the case $\alpha > 0$. By (30) and (31), we have

$$\rho_{01}^{(n)}(t) \to \langle R(\alpha t)\psi_1 | R(\alpha t)\psi_0 \rangle = \langle \psi_1 | \psi_0 \rangle, \tag{44}$$

as $n \to \infty$, and DD works.

Next, consider $\alpha < 0$. We claim that

$$\rho_{01}^{(n)}(t) \to \langle L(|\alpha|t)\psi_1 | L(|\alpha|t)\psi_0 \rangle = \int_{|\alpha|t}^{\infty} \overline{\psi_1(x)}\psi_0(x)\mathrm{d}x, \tag{45}$$

as $n \to \infty$. Since in general

$$\int_{|\alpha|t}^{\infty} \overline{\psi_1(x)}\psi_0(x)\mathrm{d}x \neq \langle \psi_1 | \psi_0 \rangle = \rho_{01}(0), \tag{46}$$

this will imply that DD does not work. If $\psi$ has support in $[|\alpha|t, \infty)$ a.e., this follows from (34). For $\psi$ with support in $(0, |\alpha|t)$ a.e., we could not find an analytic proof of this claim but at least strong numerical evidence, which has been included in the main part of this article.

## C Details on the numerics

To numerically show Eq. (11) of the main text, we simulate the DD numerics on Matlab. This is a subtle issue, because the model is by choice a very singular one, and can be sensitive to cut-off parameters and discretisation. To this end, we found that the most reliable and reasonably efficient results were achieved by using the model on the full real line and to use a discretisation of the Fourier integral. Here is the full code we used for the first example from Table I of the main text. It takes about a minute to run on a laptop

```
% Discretisation, cut-off and parameters
dx=0.01; xmax=20; n=20; t=2; alpha=-1;
% Initial state
x=(-xmax:dx:xmax)';
psi=sign(x).*abs(x).^2.*exp(-abs(x).^2/5);
```

```
psi=psi/sqrt(psi'*psi*dx);
% Discrete approximation to Fourier integral
F=exp(-1j*x*x')*dx/sqrt(2*pi);
% Equation (9) of Supplementary Material
ua=diag(exp(-1j*alpha*abs(x)))*F'...
*diag(exp(-1j*t*x.^2/(2*n)))...
    *F*diag(exp(1j*alpha*abs(x)));
u0=F'*diag(exp(1j*t*x.^2/(2*n)))*F;
% Evolution (16) of Supplementary Material
A=(ua*u0)^n;
% Evolution (17) of Supplementary Material
B=(u0*ua)^n;
% Final fidelity Equation (28) of Supplementary Material
reached=abs(psi'*B'*A*psi*dx)
```

The main difficulty with this code is that the approximated Fourier integral contains highly oscillatory terms which create artefacts [30]. A more robust method which is however much slower implements the exact solution of the model, Eq. (22). We used this to compute the convergence with respect to $n$ up to $n = 800$ in the main text, which took about a month on a decent server. To understand the convergence of the numerics with respect to the discretisation $\Delta x$ and the cut-off $|x_{\mathrm{max}}|$ see Table 2.

Table 2: Testing the convergence of the numerical approximation for the initial wave function $\propto x^2 e^{-x^2/5}$ and parameters $\alpha = 1$ and $t = 2$ at $n = 200$. We provide the percentage relative error with respect to Eq. (11) in the main text. $\infty$ is used to indicate where the numerics diverged due to errors in the high matrix power taken. We extrapolate from the numerics that at $n = 200$ the fidelity is 2.8% from the limit $n \to \infty$.

| $|x_{\mathrm{max}}|$ | $\Delta x = 0.006$ | $\Delta x = 0.003$ | $\Delta x = 0.001$ |
|---|---|---|---|
| 1.5 | 99.9 | 99.9 | 99.9 |
| 2 | 99.6 | 99.6 | 99.6 |
| 2.5 | 48.6 | 48.3 | 48.3 |
| 3 | 20.6 | 20.7 | 20.8 |
| 3.5 | 9.6 | 9.6 | 9.6 |
| 4 | 4.9 | 4.9 | 4.9 |
| 4.5 | 3.3 | 3.3 | 3.3 |
| 5 | 2.9 | 2.9 | 2.9 |
| 5.5 | $\infty$ | 2.8 | 2.8 |
| 6 | $\infty$ | 2.8 | 2.8 |
| 6.5 | $\infty$ | 2.8 | 2.8 |

```
dx=0.003; xmax=0.003; x=0:dx:xmax; y=x'; d=length(x);n=200;
psi0=y.^2.*exp(-y.^2/5);
psi0=psi0/sqrt(psi0'*psi0);
U=exp(1j*((x.^2)'*ones(d,1)'+ones(d,1)*(y.^2)')/(4*(1/n)))...
.*sin(y*x/(2*(1/n)))*dx*(-1j)/sqrt(pi*1j*(1/n));
F=diag(exp(1j*x))*U*diag(exp(-1j*x))*conj(U);
Gtilde=diag(exp(1j*x))*conj(U)*diag(exp(-1j*x))*U;
reached=abs(psi0'*Gtilde^n*F^n*psi0)
```

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
