# Peer review of "Non-Markovian noise that cannot be dynamically decoupled by periodic spin echo pulses"

_SciPost Physics, doi:SciPost Phys. 11, 027 (2021)_

## Round 3 · Referee Report · Anonymous (Referee 1) · 2020-11-8

Strengths

1) timeliness of the topic 2) mathematical rigour 3) it challenges common wisdom with a specific counterexample

Weaknesses

1) the model analysed is somehow a toy model, most likely it can be physically simulated but it does not map straightforwardly to a ready in the lab setup.

Report

Report on “Non-Markovian noise that cannot be dynamically decoupled”
by Daniel Burgarth, Paolo Facchi, Martin Fraas, Robin Hillier

The manuscript addresses the interplay between non markovian dynamics and effective quantum noise suppression by dynamical decoupling. Here non markovian dynamics is defined in terms of the existence of Zeno region characterized by a quadratic – rather then exponential -decay dynamics.
The general wisdom is that the existence of a Zeno region provides a timescale for an efficient dynamical decoupling. Here the authors provide a counterexample for which this is not true, namely the decoherence of a qubit coupled to a particle free to move in 1D. If the qubit is in the |0> state the particle is free, when the qubit is in the |1> state the particle feels a monopole potential at the origin. Such model, characterized by a non markovian decoherence, can be mapped
onto a system consisting of a qubit coupled to a real wave function defined on the positive real axis. The authors show that the Trotter expansion corresponding to the dynamical decoupling, obtained by fast spin flips, is unphysical for some range of hamiltonian parameters (crudely speaking, in the half line picture, the particle gets kicked out of its domain of existence).

The results are correct and they are, in my viewpoint of timely interest.

Requested changes

1) I found figure 2 and its caption rather obscure. Either both are made more quantitative of it would be better to drop it altogether.

---

## Round 3 · Referee Report · Anonymous (Referee 2) · 2021-2-16

Strengths

1) written in a mixed style using mathematical rigor and numerical analysis

2) part of a series of papers written by this group of authors - and contains new results

Weaknesses

1) looks like the numerical analysis has been included to make up for the lack of a solid mathematical result.

3) the counter-example is highly artificial, and the behavior displayed is not what practitioners using dynamical decoupling schemes would ever worry about. In fact, they would never consider the infinitely fast limit scheme which is the main mathematical focus here.

4) the discussion over the counter-example and its significance is convoluted.

5) No evidence that this is actually a "folklore"

6) the non-Markovian model considered here is trivially a single particle - I suspect the authors are using the term non-Markovian as it sounds flashy and suggests a stronger result than mere Markovian models. The reverse is true: Markov models require infinite-dimensional heat baths and capture the properties of realistic baths which can never be captured by the primitive model considered here.

Report

Referee report on “Non-Markovian noise that cannot be dynamically decoupled”
Authors: D. Burgarth, P. Facchi, M. Fraas, R. Hillier

Submitted to SciPost
February 2021

Dynamical decoupling is a standard technique used as a practical technique for noise suppression in physical systems. In this paper it is understood however as an idealized mathematical limit – one which cannot be realized in Nature.

The paper is one of a series by this group of authors in which they challenge conventional “physics folklore” surrounding dynamical decoupling. Unfortunately, they never point to any instances in the physics literature where this folklore is declared. Here the challenged folklore is that the so-called Zeno region provides an inherent time-scale for dynamical decoupling – but with no back up references to support this. I’m not aware of this being a concern for practitioners of dynamical decoupling.

The metier of these authors seems to be the construction of a mathematical counter-example to the perceived folklore, but frequently this will be a highly contrived model which exhibits counter-intuitive behavior in a specific and limited way, and which would never realistic arise in physics. The model of a qubit couple to a magnetic monopole considered here falls into this category.

The authors mention non-markovianity, but it is not clear what they mean by this. They seem to equate just markovianity with exponential decay. They make the emphatic claim that they "have non-Markovian noise which cannot be dynamically decoupled", but their noise model is highly artificial and it would not be a major surprise that a realistic non-Markovian environment could be constructed that does not allow their (extreme) intrepretation of dynamical decoupling.

There is also a reference to Berry’s work – but crucially no mathematical connection other than that their “numerics” shows “fractal-like” behavior. In their case the bath seems to consist of just a single particle – no one working in quantum open systems would consider such a model as a realistic source of quantum noise.

I am particularly puzzled by the following bizarre phrase in the conclusion: “While for many systems encountered in practice such reasoning is probably valid, one cannot expect to be able to prove it rigorously, because we obtained an explicit counterexample”. Surely the whole point of a counter-example is that shows immediately that a particular statement cannot be proved true! What of course they mean is that their model is an unnatural one which working physicists would dismiss as pathological.

The paper deals with themes that are not of interest to researchers working on practical dynamical coupling, and concentrates on finding unrealistic mathematical models which they claim are significant but which seem highly artificial. I cannot recommend for publication.

---

## Round 4 · Referee Report · Anonymous (Referee 1) · 2021-6-6

Strengths
I basically confirm my viewpoints as in my previous report, namely:
1) timeliness of the topic 2) mathematical rigour 3) it challenges common wisdom with a specific counterexample
Weaknesses
1) And I still think that the model analysed is somehow a toy model, most likely it can be physically simulated but it does not map straightforwardly to a ready in the lab setup.

---

## Round 4 · Referee Report · Anonymous (Referee 3) · 2021-6-22

Strengths
1) Dynamical decoupling is a timely topic. 2) The work appears to be mathematically sound. 3) The paper is easy to read.
Weaknesses
1) The example considered is artificial. 2) The title and abstract are misleading and overstate the actual results.
Report
I agree with the previous referees that the example given is highly artificial and will likely not change the course of anyone’s research, even of those directly working on dynamical decoupling, as it is far from clear whether it has any consequences for physically relevant systems. The authors point out that the half-line model can be reconstrued as a model of a particle on a full line interacting with a magnetic monopole. However, given that the existence of magnetic monopoles remains dubious, I don’t think this makes the model any more physically relevant. This being said, the result could still be of some value if it eventually leads to insights about dynamical decoupling applied to more relevant systems.
Aside from the artificial nature of the example presented, another issue that concerns me is that the title and abstract are misleading. For example, the title reads “Non-Markovian noise that cannot be dynamically decoupled”. Even if one accepts the definition of non-Markovian noise used in this work, it cannot be claimed that this noise “cannot be dynamically decoupled” based on the results presented. The authors only show that one type of dynamical decoupling fails to do the job; as far as I can tell, they do not show that all possible DD schemes will also fail for this model. I appreciate that the authors are aiming for mathematical rigor in their analysis; I would encourage them to also aim for rigor in their language.
Another statement that I found troubling is the following sentence near the top of page 3: “The example also shows, as a byproduct, that even in 1D, quantum magnetism can be interesting.” This simple and unnecessary sentence dismisses a vast literature on magnetism in one dimension. See for example this old article in Physics Today: Physics Today 31, 12, 32 (1978). The sentence should be removed.
In summary, I recommend publication after the above changes are implemented.
Requested changes
-
Change the title and abstract to better reflect the actual findings.
-
Remove the sentence about 1D magnetism on page 3.

---

## Round 4 · Referee Report · Anonymous (Referee 4) · 2021-6-26

Strengths
- The results are intriguing
- The topic is of timely interest.
- The result is surprising (at least at the first sight)
Weaknesses
- The model is pathological
Report
The effect should come essentially from the Dirichlet boundary condition, which amounts to an infinitely high potential for x<0. With such a potential, the backaction of the dynamical decoupling at higher frequencies would excite higher states in the "bath", which is evident in the wavefunctions shown in Fig. 3. One can argue that if the energy spectrum of the bath is finite or if the potential is smooth, the pathological behaviors of the the dynamical decoupling on the special model would disappear. The authors may want to comment on such an aspect.
Nonetheless, the results in this paper are interesting, in that they show the importance of backaction of dynamical decoupling and may have applications in non-pathological models. For example, to dynamically decouple an interacting bath prepared in a low-energy state, one should design the dynamical decoupling such that it would not cause the excitations of high-energy modes or the DD scheme could be spoiled.
Requested changes
Add comments on the picture of excitation of high energy states by the backaction of the fast DD.

---

## Round 4 · Author Response

The referee appears to be mostly happy with the paper. We removed Fig. 2 as requested. Regarding the statement that our model is a toy model, see our reply to Report 2.
We provide a line-by-line rebuttal of the critique in Report 2.
“Looks like the numerical analysis has been included to make up for the lack of a solid mathematical result”. We would like to point out that numerical analysis is a recognised and solid mathematical discipline. It is common practice in theoretical physics to use numerics whenever exact proofs appear to be out of reach, and that is exactly what we did: we went analytically as far as possible and then continued numerically. We also included the source code so that the sceptical reader may verify the numerics for themselves. This makes the paper mathematically complete and solid.
“The counter-example is highly artificial.” Counter-examples to folklore statements tend to be less obvious, less natural and less intuitive - otherwise the folklore would not have arisen in the first place. We agree that the statement in consideration is probably true for a number of common models but our aim here is to show that it is not always true; hence one will have to be careful when using the statement and it will make sense to determine more refined conditions in future as to when it does hold. That said, our model is a mathematical toy model that fulfills all requirements of a quantum mechanical model and it is hence sufficient to contradict the folklore statement. Whether a model is relevant or physical is a rather subjective question and the task of “finding a relevant counter-example” is not at all well-defined. We are not in any way claiming that one will find our counterexample “in general” in the lab, although we showed in the magnetic picture that the Hamiltonian shares common features with those encountered, and we hope that our discussion in Section 5 about the possible addition of local bath and system terms will make you more convinced of its physicality. The singular features of our Hamiltonian are also shared amongst most of the models used in quantum open systems. So our counterexample can stand as a warning: here’s a new feature which people were previously unaware of why DD can fail. Here’s an example which shows that one has to be careful with lots of bread-and-butter techniques in open system and control (spectral density, perturbative treatments, bath-system decomposition) when there are unbounded baths. It also should be considered as a starting point and guidance - what features (additional hypothesis) would future proofs have to have so that the folk-knowledge is restored?
“The behavior displayed is not what practitioners using dynamical decoupling schemes would ever worry about. In fact, they would never consider the infinitely fast limit scheme which is the main mathematical focus here.” Practitioners in dynamical decoupling are interested in high decoupling pulse frequencies - typically, the higher the frequency the smaller the decoupling error, and the fundamental idea of dynamical decoupling is that the decoupling error should converge to 0 as the pulse frequency tends to infinity. If decoupling does not work in this limit then it won’t work for arbitrary finite frequencies either, so a statement about the limit is actually stronger and mathematically more elegant. Also, we would like to point out that in practice when working numerically, we always deal with finite frequencies, which is exactly what would happen in the lab. Hence the behaviour we study is highly relevant.
“The discussion over the counter-example and its significance is convoluted.” The referee does not provide any justification for this statement and it is unclear what could be changed about the exposition to address this concern.
“No evidence that this is actually a "folklore".” This is not true. On p.2 of the original version we introduced the statement and provided references [13,14]. [14] is a review article. In order to strengthen this point, we added further exemplary references [15,16,17] in the revised version. We have also replaced “folklore” with the more appropriate term “mechanism” in the manuscript.
“The non-Markovian model considered here is trivially a single particle”. Notice that despite its simplicity, the Lee Friedrichs model is fully Markovian even in the single excitation sector. Moreover, our model is also infinite dimensional (although not with infinitely many modes). The referee writes that we chose a non-Markovian model due to it being “flashy”. This is incorrect. It is already known that Markovian models cannot be decoupled, so to go beyond the state of the art we had to consider non-Markovian ones. We agree with the referee that Markovian dilations tend to be mathematically much more complex than non-Markovian ones. It is exactly in this spirit that it is surprising that even simple models such as ours cannot be decoupled. The purpose of the paper is to emphasise this to the community which believes otherwise (see above).
“The authors mention non-markovianity, but it is not clear what they mean by this.” This is not true, we did define it (comment [9]). This is clearly non-Markovian according to most measures, although some authors (including ourselves in other contexts) would also consider certain models (e.g, the Shallow Pocket model) with exponential decay non-Markovian.
“I am particularly puzzled by the following bizarre phrase in the conclusion...” What we meant by our original sentence is that our model should not be classified as pathological and hence be dismissed but that there may be additional meaningful hypotheses which could be added in order to exclude counterexamples such as ours, and that this is an important future line of studies. We have now changed that sentence in order to be less contentious.

---

## Round 4 · List of Changes

Changes we made to the manuscript: * We added more references in Section 1 to show that the Zeno effect is indeed used as a mechanism to explain dynamical decoupling and have replaced “folklore” with “mechanism”.
-
We rephrased the sentence in Section 5 that Referee 2 had flagged up.
-
We removed Fig. 2 as requested by Referee 1.

---

## Round 5 · Author Response

Dear Editor,

Thank you for your message. We have implemented all minor revisions to the referees as suggested. In particular, we have amended the title and clarified the remark about magnetism in 1D as requested by Referee 2, and have added a comment about the back-action of DD along the lines of Referee 3. Referee 1 did not suggest edits. We thank the referees for helping to make our statements more precise.

Kind regards, the Authors

---

## Round 5 · List of Changes

* Title changed from "Non-Markovian noise that cannot be dynamically decoupled" to "Non-Markovian noise that cannot be dynamically decoupled by periodic spin echo pulses".

* Abstract modified accordingly, "even though the system shows a quadratic decay (a Zeno region revealing non-Markovian noise), it cannot be decoupled, no matter how fast the rotations." becomes "even though the system shows a quadratic decay (a Zeno region revealing non-Markovian noise), it cannot be decoupled by periodic spin echo pulses, no matter how fast the rotations.".

*Added "This qualitative picture might have implications for other systems, too. For example, to dynamically decouple an interacting bath prepared in a low-energy state, one should design the dynamical decoupling such that it would not cause the excitations of high-energy modes or the decoupling scheme could be spoiled." at the end of Section 4.

* Changed sentence in conclusion to "Finally our model shows that in the time-dependent case, not all dynamical effects of quantum magnetism on $L^2(\mathbb{R})$ can be removed by gauge transformations [24]" to distinguish it from one-dimensional spin physics.

---

## Editorial Decision

published